# Sublethal Exposure to Common Benzalkonium Chloride Leads to Antimicrobial Tolerance and Antibiotic Cross-Resistance in Commensal and Opportunistic Bacterial Species

Sheareazade A. Pena, Juana G. Salas, Nilisha Gautam, Ashley M. Ramos and Aubrey L. Frantz *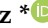

Department of Natural Sciences, University of North Texas at Dallas, Dallas, TX 75241, USA
* Correspondence: aubrey.frantz@untdallas.edu

**Abstract:** The production and consumer use of disinfectants has substantially increased during the COVID-19 pandemic. Benzalkonium chloride (BAC) is a mixture of alkyl benzyl dimethyl ammonium chloride compounds and is the most common active ingredient in surface cleaning and disinfecting products. Accordingly, BAC compounds are routinely in contact with microorganisms in indoor environments, which may contribute to the development of antimicrobial tolerance and cross-resistance. To investigate the impact of BAC exposure on commensal and opportunistic bacteria of public health importance, we exposed *Staphylococcus epidermidis*, *Corynebacterium xerosis*, *Staphylococcus aureus*, *Klebsiella pneumoniae*, *Escherichia coli*, and *Pseudomonas aeruginosa* to a standard BAC mixture ($BAC_{12-14}$), as well as purified $BAC_{16}$. Minimum inhibitory concentrations (MICs) and antibiotic susceptibilities were determined before and after repeated exposure to sublethal BAC concentrations. MICs for Gram-negative bacteria were significantly higher than Gram-positive bacteria. Additionally, $BAC_{12-14}$ MICs were significantly higher for opportunistic pathogens and BAC-tolerance was associated with antibiotic cross-resistance. These results suggest that common Gram-negative opportunistic pathogens are less sensitive to BAC-inhibition than commensal species and may preferentially develop antimicrobial tolerance upon repeated or prolonged exposure to $BAC_{12-14}$. Reevaluating the formulation and concentration of BAC-containing products in efforts to limit the development of antimicrobial tolerance and antibiotic co-resistance is warranted.

**Keywords:** disinfectants; antimicrobial resistance; benzalkonium chloride; opportunistic pathogens; commensal bacteria; quaternary ammonium compounds; COVID-19

## 1. Introduction

Benzalkonium chloride (BAC) compounds are a distinct group of quaternary ammonium compounds (QACs) with broad-spectrum antimicrobial activity [1]. Accordingly, these compounds are frequently used as active ingredients in many antimicrobial, hygiene and pharmaceutical products. BACs are generally considered membrane-active antimicrobial compounds that cause membrane damage, microbial cell leakage and subsequent cell or particle lysis in bacteria, yeast and certain viruses [1]. BACs are approved for a variety of functional uses and are commonly used in disinfectants, antiseptics, sanitizers and numerous household cleaning and personal care products [2]. These compounds are appealing as disinfectants and cleaning agents because they are relatively stable, fast-acting and remain biologically active on surfaces for extended lengths of time, even hours after application [3]. BAC concentrations are typically 0.01–0.05% for BAC-containing products applied directly to the skin, and 0.1–0.5% in surface disinfectants and cleaning products [4]. Since these products are generally not rinsed off after application, the BAC compounds are expected to remain active on the skin as the product dries.

In the United States, the EPA is responsible for registering antimicrobial pesticides, including QACs. There are approximately 10,000 registered QAC formulations and the

EPA has clustered QACs into four structural groups for testing purposes [5]. BAC was the first QAC registered in the U.S. in 1947, quickly designated a high production volume chemical, and has since been one of the longest-used active ingredients in disinfectants and cleaning products [6]. During the production of BAC, mixtures of BAC compounds with even-numbered alkyl chain lengths of 8–18 carbons ($BAC_8$–$BAC_{18}$) are produced. The most common product formulations on the market include various combinations of $BAC_{12}$, $BAC_{14}$ and $BAC_{16}$ [3]. Accordingly, BAC with an alkyl chain length of 12–16 carbons ($BAC_{12-16}$) has been identified as the model Group II QAC (non-halogenated benzyl substituted QAC) with over 300 antimicrobial product registrations [3]. Despite the numerous BAC-containing products on the market, product labels typically only identify BAC as an active ingredient and do not provide the formulation of BAC compounds. Likewise, a notable limitation to previous toxicological and efficacy data is that these assessments are largely unpublished company or regulatory reports that are conducted on individual QACs or QAC subgroups (e.g., $BAC_{12-16}$), yet most commercial QAC-containing products contain mixtures of QACs and compounds [3,7]. Therefore, under these guidelines, all $BAC_{12-16}$ mixtures are assessed together, regardless of composition. It should also be considered that, while toxicity assessments generally assume additive effects of compounds, available data suggests that many toxicant mixtures, including QACs, do not demonstrate additivity [8–10]. These concerns highlight the importance of identifying the precise composition and evaluating the efficacy and toxicity of the BAC mixture.

In 2020, the CDC included QACs on the EPA's List N Disinfectants, which identifies products that meet EPA criteria for use against SARS-CoV-2 [11,12]. Of the more than 500 products on the EPA's list of recommended disinfectants, nearly half contain BAC as the active ingredient. Consequently, the production of BAC-containing products has significantly risen during the COVID-19 pandemic and is forecasted to continue to increase in the years to come [6]. Several COVID-19 pandemic-related studies have shown that disinfectants are often not used as directed on their product labels and there are numerous factors that can reduce QAC efficacy [13–15]. Considering the prolific use of BAC-containing products and the stability of these compounds post-application, it is predicted that bacteria are likely exposed to an effective concentration of QACs at the time of initial application and to sublethal concentrations through secondary and repeated exposures [7,16,17]. Prolonged exposure to sublethal concentrations of QACs may select for QAC-tolerant organisms and therefore simultaneously increase the abundance of antimicrobial resistant organisms [18–20]. In 2019, WHO identified antimicrobial resistance as one of the top ten global public health threats that require urgent multisectoral action, and in 2022, the EU elevated the concern of antimicrobial resistance to the top three priority health threats [21,22]. Due to the widespread use during the COVID-19 pandemic, numerous studies have identified QAC disinfectants as an emerging and leading cause of bacterial resistance to disinfectants and antibiotics [6,23,24]. Pre-pandemic studies that have investigated bacterial adaptations to QACs in vitro have predominantly used an unspecified BAC composition and concentration, and were limited to select pathogenic or environmental bacterial strains [16,17,20,25–28]. Here, we determined and compared antimicrobial activities of purified $BAC_{16}$ and a standard BAC mixture ($BAC_{12-14}$) against common commensal and opportunistic bacterial species. $BAC_{12-14}$ was purchased from Sigma-Aldrich and is composed of approximately 70% $BAC_{12}$, 30% $BAC_{14}$, and trace amounts of $BAC_{16}$. We employed a realistic scenario of bacterial acclimation to evaluate the development of BAC tolerance following repeated exposure to sublethal concentrations of BAC compounds and assessed the development of antibiotic co-resistance.

## 2. Materials and Methods

### 2.1. Bacterial Species and Growth Conditions

Bacterial species selected for this study include *Staphylococcus epidermidis*, *Corynebacterium xerosis*, *Staphylococcus aureus*, *Klebsiella pneumoniae*, *Escherichia coli* (0157:H7), *Pseudomonas aeruginosa* and *Escherichia coli* ATCC 25922. All bacteria were purchased from

Carolina Biological Supply Company (MicroKwik Cultures® or Living Tubes, Burlington, NJ, USA) and reconstituted based on manufacturer instructions.

## 2.2. $BAC_{12-14}$ and Purification and Identification of $BAC_{16}$

$BAC_{12-14}$ was purchased from Sigma-Aldrich (B6295; CAS-no 63449-41-2, St. Louis, MI, USA). $BAC_{16}$ was purchased from Sigma-Aldrich (B4136; CAS-no 122-18-9, St. Louis, MI, USA) and came in a powdered form. $BAC_{16}$ was further purified by recrystallization from ethanol (95% ethanol solution at 253 K), as previously described [29]. The identity of the pure compound was characterized and confirmed by mass spectrometry and crystal structure analysis [29]. The analyses were performed at the University of Texas at Arlington using a Shimadzu GCMS-QP2010 SE (Canby, OR, USA) for mass spectrometry and a Bruker SMART Apex II X-ray Diffractometer (Madison, WI, USA) for crystal structure analysis.

## 2.3. Determination of BAC Minimum Inhibitory Concentrations (MIC) and Minimum Bactericidal Concentrations (MBC)

MICs and MBCs were determined by microdilution according to Clinical and Laboratory Standards Institute (CLSI) standards [30]. BAC concentrations ranged between 0.1 mg/L and 2000 mg/L. Growth was determined through turbidity analysis. MICs were determined to be the lowest concentration of BAC compound that prevents visible growth of the bacteria. MBCs were determined by transferring 10 μL aliquots from wells exhibiting no turbidity (optically clear wells) to sterile media prior to 4 days of incubation at 37 °C. The MBC was determined to be the lowest concentration of BAC at which no growth occurred after transfer and 4 days of incubation.

## 2.4. Bacterial Acclimation to BAC Compounds

Overnight cultures of bacteria were diluted 1:100 into the BAC-containing growth medium (sublethal concentrations 0.1 mg/L–15 mg/L of BAC) and incubated at 37 °C aerobically for 24 h. Bacteria were passaged up to 10 times in the presence of the sublethal BAC concentrations while using a consistent inoculation density for each passage ($1 \times 10^6$ CFU/mL). The sublethal concentrations used were 60–80% of the P0 MICs. Cultures were plated between all passages to check for purity and viability. MICs for a minimum of eight bacterial isolates for each species were determined, which included a minimum of two independent experiments per isolate, with three replicates per experiment. Mean MICs were determined by averaging MIC values from all isolates within each species. The progenitor strains that were not passaged or exposed to BAC is designated P0. Progenitor strains that were passaged alongside experimental groups, but were not exposed to sublethal BAC concentrations, were used as controls (P10-control). MICs and MBCs were determined (as described above) every 10 passages. P10 control MICs were not statistically different from P0 MICs (Supplementary Table S1).

## 2.5. Determination of the Contribution of Efflux Pump Activity to Antimicrobial Tolerance

The activities of bacterial efflux pumps were evaluated using the efflux pump inhibitor, phenylalanine-arginine β-naphthylamide (PAβN). For BAC-tolerant strains, MIC assays were repeated, as described above, in the presence or absence of 25 μg/mL of PAβN.

## 2.6. Determination of Antibiotic Susceptibilities

Antibiotic susceptibility tests were performed using the disc diffusion method, according to the CLSI guidelines [30], before and after repeated exposure to sublethal concentrations of BAC compounds. Briefly, overnight bacterial cultures were diluted and inoculated on Müller–Hinton agar. Antibiotic susceptibility discs (30 μg neomycin, 10 μg bacitracin, 10 μg penicillin, 30 μg chloramphenicol and 10 μg streptomycin) were purchased from Carolina Biological Supply Company. Individual antibiotic discs were placed on inoculated plates and the diameter of the inhibition zones were measured after incubation at 37 °C for 24 h. Data were reported as susceptible (S), intermediate (I) or resistant (R), based on

CLSI guidelines [30]. Per CLSI guidelines, *Escherichia coli* ATCC 25922 was used as a quality control strain.

### 2.7. Statistical Analyses

Individual MIC measurements for bacterial isolates before (P0) and after (P10) repeated BAC exposure and EPI treatment were grouped and subjected to two-tailed *t*-test analyses. One-way ANOVA, Levene's test to assess equality of variance and Welch F tests to address unequal variance were performed to compare MICs across all bacterial species and between $BAC_{12-14}$ and $BAC_{16}$. All calculations were performed using the Data Analysis Toolpak in Microsoft Excel (Microsoft 365). Means ± standard deviations and *p* values were reported (* *p* < 0.05), each of which includes measurements from two independent experiments, with three replicates per experiment, for a minimum of eight bacterial isolates for each species.

## 3. Results

### 3.1. BAC Compounds and Purification of $BAC_{16}$

Benzalkonium chloride (BAC) is a mixture of alkyl benzyl dimethyl ammonium chlorides with various even-numbered alkyl chain lengths. $BAC_{12-14}$ was obtained from Sigma-Aldrich and is composed of approximately 70% $BAC_{12}$, 30% $BAC_{14}$, and trace amounts of $BAC_{16}$. $BAC_{16}$ was also obtained from Sigma-Aldrich, further purified by recrystallization from aqueous ethanol and characterized and confirmed by mass spectrometry and crystal structure analysis (Supplementary Figure S1) [29]. This process produced the high purity compound that was used in this study.

### 3.2. Bacterial Sensitivities to BAC-Inhibition

To assess the antimicrobial activity of the BAC compounds, we determined the MICs for $BAC_{12-14}$ and $BAC_{16}$ against selected commensal and opportunistic bacterial species. The bacteria selected to use in this study include *Staphylococcus epidermidis*, a common Gram-positive skin commensal species; *Corynebacterium xerosis*, a common Gram-positive commensal species of the skin and nasopharynx; *Staphylococcus aureus*, a leading opportunistic pathogen and the standard Gram-positive test organism for QAC efficacy testing; *Klebsiella pneumoniae*, a Gram-negative, opportunistic pathogen and a leading cause of nosocomial infections; *Escherichia coli*, a common, Gram-negative, opportunistic gastrointestinal pathogen; and *Pseudomonas aeruginosa*, a common, Gram-negative, opportunistic pathogen that is found widely in the environment. Mean MICs following initial BAC exposure are identified as P0 MICs and found in Table 1. P0 MICs ranged between 0.20–25 mg/L. These values are consistent with reported MICs for other BAC compounds and BAC containing products, and generally lower than directed use concentrations for commercial products [1,25].

**Table 1.** P0 MICs for $BAC_{12-14}$ and $BAC_{16}$ compounds.

| MIC (mg/L) | $BAC_{12-14}$ | $BAC_{16}$ |
|---|---|---|
| *S. epidermidis* | 0.24 ± 0.03 | 0.24 ± 0.02 |
| *C. xerosis* | 0.58 ± 0.12 | 0.40 ± 0.13 |
| *S. aureus* | 0.63 ± 0.18 | 0.28 ± 0.08 |
| *K. pneumoniae* | 2.69 ± 0.43 | 0.56 ± 0.11 |
| *E. coli* | 4.50 ± 0.50 | 1.94 ± 0.17 |
| *P. aeruginosa* | 13.75 ± 2.17 | 20.00 ± 2.50 |

MICs are shown as means ± standard deviations. Mean values include a minimum of two independent experiments with three replicates per experiment, for eight bacterial isolates per species.

Bacterial sensitivity to BAC-inhibition was observed to be BAC-dependent. The differences between average $BAC_{12-14}$ MICs and average $BAC_{16}$ MICs for each bacterial species were statistically significant (*p* < 0.05), with the exception of *S. epidermidis*. Average P0 $BAC_{12-14}$ and $BAC_{16}$ MICs for *S. epidermidis* were not statistically different and were the

lowest average MICs, among all tested bacterial species (Figure 1 and Table S2). Likewise, MICs for Gram-negative bacteria were significantly higher than MICs for Gram-positive bacteria, indicating that these Gram-negative opportunistic pathogens were less sensitive to BAC-inhibition. These trends were also seen when evaluating $BAC_{12-14}$ and $BAC_{16}$ minimum bactericidal concentrations (MBCs) (Tables S3 and S4). Overall, species-specific sensitivity to BAC-inhibition was observed for $BAC_{12-14}$, as P0 MICs were significantly different across almost all bacterial species (Table 1, Tables S5 and S6)–the only exception were the $BAC_{12-14}$ MICs for *S. aureus* and *C. xerosis*. In contrast, while P0 $BAC_{16}$ MICs were statistically different among the Gram-negative species, P0 $BAC_{16}$ MICs were not significantly different among the Gram-positive species (Table S6). Of note, *S. aureus*, the standard test organism of QAC efficacy testing, was significantly more sensitive to BAC-inhibition than *K. pneumoniae*, *E. coli* and *P. aeruginosa*. In particular, the average $BAC_{16}$ MIC for *S. aureus* was more than 70-fold less than that of the opportunistic pathogen, *P. aeruginosa*. Likewise, common commensal species, *S. epidermidis* and *C. xerosis*, were significantly more sensitive to BAC-inhibition than any of the Gram-negative bacterial species.

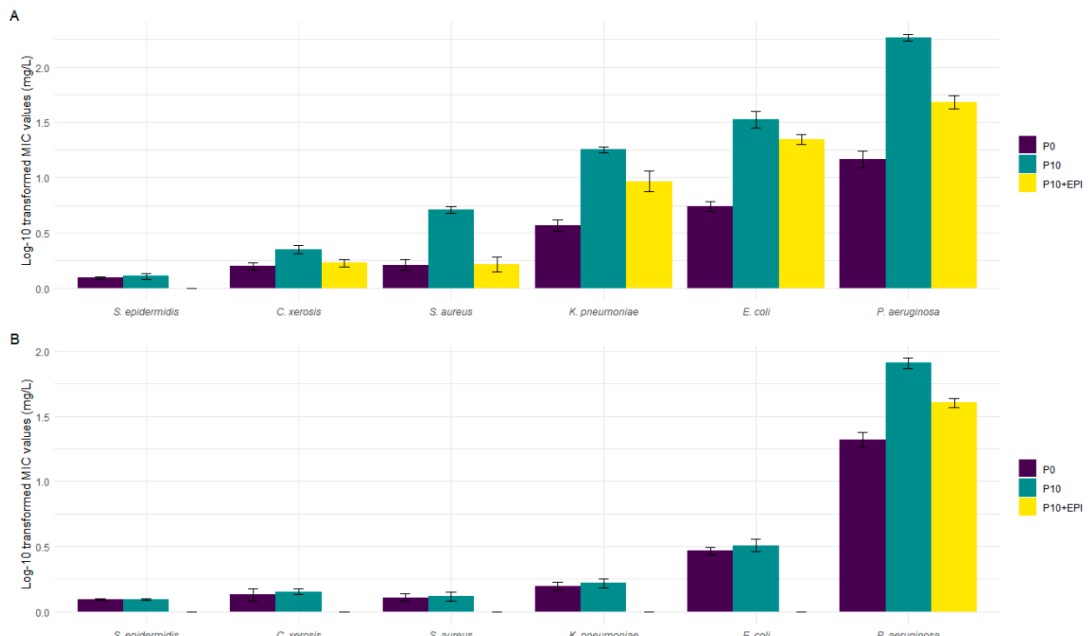

**Figure 1.** MICs before and after repeated exposure to $BAC_{12-14}$ and $BAC_{16}$. (**A**) $BAC_{12-14}$ log-10 transformed mean MICs. (**B**) $BAC_{16}$ log-10 transformed mean MICs. Log-10 transformed mean MICs are graphed with error bars representing standard deviations. P0, initial exposure to BAC compounds; P10, bacteria were cultured in sublethal concentrations of BAC compounds for 10 consecutive passages; P10+EPI, BAC-acclimated bacteria were treated with an efflux pump inhibitor, PaβN.

### 3.3. Repeated Exposure to Sublethal BAC Concentrations Leads to Antimicrobial Tolerance in Opportunistic Pathogens

Various factors are known to affect QAC efficacy, including temperature, contact time, type of surface treated and the method of application [7]. As a result, in practice, BAC-containing products are often applied at less than directed use concentrations [25]. To determine if bacteria would develop tolerance to BAC compounds, bacteria were exposed to sublethal $BAC_{12-14}$ and $BAC_{16}$ concentrations for 10 consecutive passages (P10). BAC concentrations of 60–80% P0 MICs were used as sublethal doses to mimic realistic scenarios of BAC application. Mean MIC values following 10 passages in the presence of sublethal BAC concentrations are identified as P10 MICs and are presented in Table 2. Comparison between P0 and P10 MICs across species can be seen in Figure 1. P10 MICs ranged from 0.2–200 mg/L and were, on average, more than 10-fold greater than P0 MICs for some bacterial species. Interestingly, the only species that developed tolerance to $BAC_{16}$

was *P. aeruginosa*. No statistically significant differences between $BAC_{16}$ P0 and P10 MICs were observed for *S. epidermidis*, *C. xerosis*, *S. aureus*, *K. pneumonia* or *E. coli*. In contrast, all bacterial species, except *S. epidermidis*, developed tolerance to $BAC_{12-14}$, as demonstrated by the statistically significant increase in mean P10 MIC values, compared to P0 values (Table 2). The mean $BAC_{12-14}$ P10 MIC for *P. aeruginosa* was more than 13-fold greater than the mean P0 MIC, as compared to only a 4-fold increase following repeated exposure to $BAC_{16}$, and was more than 650-fold greater than the mean $BAC_{12-14}$ P10 MIC for *S. epidermidis*. Taken together, the species-specific differences in observed BAC sensitivities were amplified as a result of repeated exposure to sublethal $BAC_{12-14}$ concentrations.

**Table 2.** P10 MICs before and after treatment with the efflux pump inhibitor (EPI), PaβN.

| MIC (mg/L) | $BAC_{12-14}$ | | $BAC_{16}$ | |
|---|---|---|---|---|
| | **P10** | **P10 + EPI** | **P10** | **P10 + EPI** |
| *S. epidermidis* | $0.28 \pm 0.10$ | NT | $0.24 \pm 0.20$ | NT |
| *C. xerosis* | $1.25 \pm 0.20$ * | $0.69 \pm 0.10$ | $0.40 \pm 0.10$ | NT |
| *S. aureus* | $4.13 \pm 0.30$ * | $0.66 \pm 0.20$ ** | $0.31 \pm 0.11$ | NT |
| *K. pneumoniae* | $17.00 \pm 1.00$ * | $8.44 \pm 1.70$ **# | $0.66 \pm 0.12$ | NT |
| *E. coli* | $33.13 \pm 5.60$ * | $21.25 \pm 2.20$ **# | $2.25 \pm 0.35$ | NT |
| *P. aeruginosa* | $184.38 \pm 12.1$ * | $47.50 \pm 6.60$ **# | $80.63 \pm 7.26$ * | $39.38 \pm 3.00$ **# |

MICs are shown as means $\pm$ standard deviations. Mean values include a minimum of two independent experiments with three replicates per experiment, for eight bacterial isolates per species. +EPI, P10 MIC assays were repeated for BAC-acclimated bacteria in the presence of 25 μg/mL of PaβN; NT, Not Tested—P10 MICs were not statistically different than P0 MICs in BAC-acclimated bacteria.* P10 MIC is significantly different than the P0 MIC ($p < 0.05$). ** P10 MIC in the presence of the EPI is significantly different than the P10 MIC in the absence of the EPI ($p < 0.05$). # P10 MIC in the presence of the EPI is significantly different than the P0 MIC ($p < 0.05$).

### 3.4. Treatment with an Efflux Pump Inhibitor (EPI) Reduces Tolerance and Restores Sensitivity to BAC-Inhibition

Previous studies have indicated that the expression of multidrug efflux pumps is largely responsible for biocide and antibiotic resistance [31–34]. Efflux systems with broad specificity are observed in a vast number of bacteria, and function to pump the antimicrobial substance out of the cell, thus, reducing the intracellular concentrations of biocide and increasing antimicrobial tolerance. Particularly, the plasmid-encoded efflux pumps of the Qac protein family and the chromosomally encoded NorA and NorB multidrug efflux pumps are widely distributed in the commensal and pathogenic bacterial species used in this study, as well as in other clinically important bacterial species [19,31,33,35]. To determine if the observed experimental increases in MICs following repeated exposure to sublethal BAC concentrations were due to the activity of efflux pumps, we repeated the MIC assays for BAC-acclimated bacteria, in the presence or absence of the EPI, PaβN. Treatment with PaβN significantly increased BAC sensitivity and reduced P10 MICs in all BAC-tolerant bacteria (Table 2 and Table S7, Figure 1). Specifically, treatment with PaβN restored BAC sensitivity in $BAC_{12-14}$-acclimated *C. xerosis* and *S. aureus*–P10 + EPI MICs were not statistically different than P0 MICs. However, while treatment with PaβN significantly reduced P10 MICs for $BAC_{12-14}$-acclimated *E. coli*, *K. pneumoniae* and *P. aeruginosa*, as well as $BAC_{16}$-acclimated *P. aeruginosa*, P10 + EPI MICs were statistically different than P0 values (Table 2 and Table S7). These results strongly suggest that the observed BAC tolerance was mediated, at least in part, by bacterial efflux pump activity. Recent evidence has shown that exposure to QAC compounds can lead to the proliferation of antibiotic resistance genes [36,37] and EPIs have recently become an attractive potential therapeutic strategy to combat the rise in antimicrobial resistance [31]. This is particularly true for resistant staphylococci strains in healthcare settings [27,38–40]. Experts have suggested that the use of efflux pump inhibitors in combination with QACs may be beneficial in slowing the rate of resistance and prolonging the lifespan of these products [24,41,42]. These preliminary results support continued research into the therapeutic and strategic

employment of efflux pump inhibitors, although further investigation into mechanisms of BAC-tolerance is necessary.

### 3.5. BAC Tolerance Is Associated with Antibiotic Cross-Tolerance in BAC-Acclimated Bacteria

To determine if tolerance to BAC compounds could co-select for antibiotic cross-resistance, we assessed antibiotic susceptibilities in bacterial isolates before and after repeated exposure to sublethal concentrations of $BAC_{12-14}$ and $BAC_{16}$. Baseline (P0) antibiotic susceptibilities for each bacterial species against neomycin, bacitracin, ciprofloxacin, chloramphenicol, streptomycin and penicillin are found in Table 3. Antibiotic susceptibility changes were only observed in bacteria that developed BAC-tolerance following repeated sublethal BAC exposure. No changes in antibiotic susceptibilities were observed in bacteria that did not develop BAC-tolerance following repeated sublethal exposure. Accordingly, all bacteria, with the exception of *S. epidermidis*, developed cross-tolerance to at least one antibiotic. Specifically, $BAC_{12-14}$-acclimated *C. xerosis* were less sensitive to neomycin and bacitracin; $BAC_{12-14}$-acclimated *S. aureus* were less sensitive to penicillin; BAC-acclimated *K. pneumoniae* were less sensitive to chloramphenicol and streptomycin; $BAC_{12-14}$-acclimated *E. coli* were less sensitive to neomycin and $BAC_{12-14}$-acclimated *P. aeruginosa*, as well as $BAC_{16}$-acclimated *P. aeruginosa*, were less sensitive to streptomycin. Overall, changes in antibiotic susceptibilities were associated with BAC-tolerance and $BAC_{12-14}$-acclimated bacteria were more likely to develop antibiotic cross-tolerance, as compared to $BAC_{16}$-acclimated bacteria.

**Table 3.** Antibiotic susceptibility before and after repeated exposure to sublethal concentrations of BAC compounds.

| | Neomycin | Bacitracin | Chloramphenicol | Streptomycin | Penicillin |
|---|---|---|---|---|---|
| *S. epidermidis* | | | | | |
| Baseline (P0) | S | I | S | S | I |
| $BAC_{12-14}$ (P10) | S | I | S | S | I |
| $BAC_{16}$ (P10) | S | I | S | S | I |
| *C. xerosis* | | | | | |
| Baseline (P0) | S | S | S | S | R |
| $BAC_{12-14}$ (P10) | I * | I * | S | S | R |
| $BAC_{16}$ (P10) | S | S | S | S | R |
| *S. aureus* | | | | | |
| Baseline (P0) | S | I | S | S | S |
| $BAC_{12-14}$ (P10) | S | I | S | S | I * |
| $BAC_{16}$ (P10) | S | I | S | S | S |
| *K. pneumoniae* | | | | | |
| Baseline (P0) | S | R | S | S | R |
| $BAC_{12-14}$ (P10) | S | R | I * | I * | R |
| $BAC_{16}$ (P10) | S | R | S | S | R |
| *E. coli* | | | | | |
| Baseline (P0) | S | R | S | S | R |
| $BAC_{12-14}$ (P10) | I * | R | S | S | R |
| $BAC_{16}$ (P10) | S | R | S | S | R |

**Table 3.** *Cont.*

| | Neomycin | Bacitracin | Chloramphenicol | Streptomycin | Penicillin |
|---|---|---|---|---|---|
| *P. aeruginosa* | | | | | |
| Baseline (P0) | R | R | R | I | R |
| $BAC_{12-14}$ (P10) | R | R | R | R * | R |
| $BAC_{16}$ (P10) | R | R | R | R * | R |

Data are reported as susceptible (S), intermediate (I), or resistant (R). P0, initial exposure to BAC compounds; P10, bacteria were exposed to sublethal concentrations of BAC compounds for 10 consecutive passages. A minimum of three bacterial isolates were tested for each species. Antibiotic susceptibility assays were done in triplicate and susceptibility determinations did not vary between replicates. * Denotes that antibiotic susceptibility was categorically different than baseline antibiotic susceptibility (P0).

## 4. Discussion

BACs are the most common active ingredient in surface cleaning and disinfecting products, and are widely used in healthcare settings, food industries, schools, workplaces and homes [25]. A 1978 study by Richards and Mizrahi found that BAC solutions obtained from various manufacturers had significantly different antimicrobial activities, which were related to the mixture compositions [43]. Yet, even decades later, the compositions of BAC mixtures are rarely reported in peer-reviewed publications or product labels [28,44,45]. Similar to our work, numerous other studies have used BAC purchased from Sigma-Aldrich to investigate the development of BAC tolerance in various settings, yet the composition of the BAC mixture was not specified [46–48]. Kim et al. (2018) also used BAC purchased from Sigma-Aldrich to assess BAC tolerance in bacteria isolated from river sediment, although this BAC batch was reported to be a 60:40 mixture of $BAC_{12}$ and $BAC_{14}$, respectively [36]. Furthermore, available regulatory studies typically identify 40:50:10 mixture of C12:C14:C16 for toxicity assessments [3]. Results presented here support the claims that BAC compounds have different antimicrobial activities and suggest that the composition of the BAC mixture significantly affects bacterial sensitivities as well as the development of antimicrobial tolerance and cross-resistance.

While previous studies have demonstrated the ability of pathogenic bacteria to develop tolerance to BAC [25,46,47], this work is the first, to our knowledge, to compare BAC sensitivities in common commensal and opportunistic bacterial species. Our results indicate that the Gram-positive commensal bacteria were significantly more sensitive to BAC-inhibition and less likely to develop tolerance to BAC compounds, as compared to the Gram-negative opportunistic pathogens. Recent studies assessing effectiveness of disinfectants versus plain soap sanitation approaches found that the use of chemical disinfectants strongly promoted the survival of pathogenic bacteria on cleaned surfaces [49]. Additionally, environments with the highest levels of disinfectant use have been found to harbor the highest rates of multi-drug resistant microorganisms [50,51]. This is particularly relevant for high-touch hospital surfaces that are frequently treated with disinfectants, and where nosocomial infections are a principal threat [44]. The results presented here support the idea that sublethal concentrations of disinfectants could select for less sensitive opportunistic bacteria, favoring their survival and persistence in the indoor environment.

Although preliminary, these results have significant human health and public health importance. While QACs were widely used prior to the COVID-19 pandemic, production and demand for QAC-containing cleaning and disinfectant products have substantially increased since 2020 [6]. Thus, due to the prolific use of QAC products and the stable nature of BAC compounds, commensal and opportunistic bacterial species are frequently in contact with sublethal concentrations of BACs. Nearly 10% of bacterial samples isolated from local indoor community settings frequently disinfected with BAC-containing products were found to be resistant to in-use BAC concentrations [52]. QAC-acclimated bacteria can easily be transferred via dermal contact with hard surfaces, post-product application [53–55]. Survival of MRSA on automated teller machines was suggested to be facilitated by low disinfec-

tant concentrations, and numerous outbreaks have been documented and attributed to disinfectant-resistant pathogens that have contaminated antiseptic products [35,56–58]. The results presented here support these findings and indicate that prolonged exposure to sublethal concentrations of BAC can lead to the development of QAC tolerance and antibiotic cross-resistance.

Our results presented here suggest that bacterial responses and adaptations to prolonged BAC exposure vary depending on the BAC compound. Opportunistic bacterial species, *S. aureus*, *K. pneumoniae*, *E. coli* and *P. aeruginosa*, preferentially developed BAC tolerance and antibiotic cross-tolerance to $BAC_{12-14}$, as compared to pure $BAC_{16}$. These findings are alarming, as several recent studies have identified $BAC_{12}$ and $BAC_{14}$ as the most abundant QAC compounds in indoor environments and associated with adverse health effects. Zheng et al. (2020) found that QAC concentrations were significantly higher in residential dust collected during the pandemic (2020), as compared to before the pandemic (2019) [53]. QACs were detected in more than 90% of dust samples, with the greatest increases in $BAC_{12}$ and $BAC_{14}$ abundance [53]. Zheng et al. (2021) compared blood samples from 222 individuals, before and during the COVID-19 pandemic (in 2019 and 2020, respectively), and found that the total blood QAC concentration was greater than two-fold higher in samples collected in 2020 [59]. The most common QAC detected was $BAC_{12}$, which was found in more than 94% of pandemic blood samples. Importantly, QAC bioaccumulation potentials were determined and demonstrated that C12 homologs exhibited the slowest clearance rates, indicating that these QACs could preferentially bioaccumulate [59]. A 2021 study by Hrubec et al. found that more than 80% of sampled individuals contained QACs in their blood, with preferential accumulation of $BAC_{12}$ and $BAC_{14}$. These blood QAC concentrations were also correlated with biomarkers of inflammation, mitochondrial dysfunction and sterol imbalance [8]. Lastly, Zheng et al. (2022) recently investigated the QAC concentration in breastmilk. Even though less than half of the 48 mothers in the study reported that they regularly used QAC-containing disinfectant products in their homes, QACs were detected in all breastmilk samples [60]. Consistent with previous studies, $BAC_{12}$ and $BAC_{14}$ were the most abundant QAC detected in breastmilk. These results contribute to the growing evidence that formulation and composition of BAC-containing products should be a priority and potentially eliminating or reducing the use of particular QAC species may be warranted.

## 5. Conclusions

The production and consumer use of BAC-containing products have risen drastically during the COVID-19 pandemic and is likely to continue to increase over the next decade. Cumulating data indicate that BAC compounds, specifically $BAC_{12}$ and $BAC_{14}$, are abundant in our built environments and associated with adverse health and environmental effects. This work suggests that common Gram-negative opportunistic bacteria may be less sensitive to BAC-inhibition than common commensal species and may preferentially develop antimicrobial tolerance upon repeated or prolonged sublethal exposure to $BAC_{12}$ and $BAC_{14}$. Reevaluating the formulation and concentration of BAC mixtures in efforts to limit the development of antimicrobial tolerance and antibiotic co-resistance is recommended. Continued investigation is needed to guide formulation, selection and use of BAC-containing products and provide insights into the potential impact of widespread QAC use.

**Supplementary Materials:** The following supporting information can be downloaded at: https://www.mdpi.com/article/10.3390/applmicrobiol3020041/s1, Figure S1: Three-dimensional structure of $BAC_{16}$; Table S1: P10-Control Strain MICs for $BAC_{12-14}$ and $BAC_{16}$; Table S2: Individual comparisons of $BAC_{12-14}$ and $BAC_{16}$ P0 and P10 MICs; Table S3: Minimum Bactericidal Concentrations (MBCs) for $BAC_{12-14}$ and $BAC_{16}$; Table S4: Comparison of MBCs across all bacterial species; Table S5: Comparison of P0 MICs and P10 MICs across all bacterial species; Table S6: Individual comparisons of MICs between bacterial species; Table S7: Comparison of P10+EPI MICs to P0 MICs and P10 MICs.

**Author Contributions:** Conceptualization, A.L.F.; methodology, A.L.F.; investigation, S.A.P., J.G.S., N.G., A.M.R. and A.L.F.; resources, A.L.F.; data curation, A.L.F.; writing—original draft preparation, A.L.F.; writing—review and editing, A.L.F.; visualization, A.L.F.; supervision, A.L.F.; project administration, A.L.F. All authors have read and agreed to the published version of the manuscript.

**Funding:** This research received no external funding.

**Data Availability Statement:** The data presented in this study are available in the article and supplementary material.

**Acknowledgments:** We are grateful for Muhammed Yousufuddin and his research group for providing the purified BAC$_{16}$, and Nicholas Lorusso for his assistance with data visualization.

**Conflicts of Interest:** The authors declare no conflict of interest.

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
