# Peer review of "Sublethal Exposure to Common Benzalkonium Chloride Leads to Antimicrobial Tolerance and Antibiotic Cross-Resistance in Commensal and Opportunistic Bacterial Species"

_2673-8007, doi:10.3390/applmicrobiol3020041_

Round 1
Reviewer 1 Report
The manuscript presents an interesting research regarding the emergence of antibiotic tolerance and antibiotic cross-resistance after exposure of different bacterial strains to BAC.
Some minor considerations should be addressed:
- the title and keywords do not mention that commensal bacterial species were also taken into consideration
- in section 2.3, lines 117-119, the number of days of incubation should be verified and modified accordingly
- in section 2.4, line 134, data that are not shown in the manuscript should be included in the supplementary files.
Reviewer 2 Report
In this study, the authors have investigated the effect of Benzalkonium chloride (BAC) exposure on commensal and opportunistic bacteria of public health importance. They determined minimum inhibitory concentrations (MICs) for different antibiotics before and after exposure to BAC. The data suggest that the MIC concentrations were higher in bacteria upon exposure to BAC. The study is interesting, and it provides valuable information to reevaluate the formulation and concentration of BAC-containing products in efforts to limit the development of antimicrobial tolerance. The study is well-controlled, and the manuscript is well organized. I have some minor comments (given below) to improve the quality of the manuscript.
Comments:
- Use italics for scientific names throughout the manuscript
- L116: 10μl à 10 μl
- Authors mention that the expression of multidrug efflux pumps is largely responsible for biocide resistance. It was noted that the treatment with efflux pump inhibitors reduced the BAC tolerance and restored sensitivity to BAC-inhibition. Any idea which efflux pumps are involved in BAC tolerance? It would be better to mention those efflux pumps (if reported) as well.
